SCIENTIFIC CORRESPONDENCE

# Comment on 'Accumbens cholinergic interneurons dynamically promote dopamine release and enable motivation'

**James Taniguchi[1†], Riccardo Melani[1†], Lynne Chantranupong[2], Michelle J Wen[2], Ali Mohebi[3], Joshua D Berke[3], Bernardo L Sabatini[2], Nicolas X Tritsch[1]\***

[1]Neuroscience Institute and Fresco Institute for Parkinson's and Movement Disorders, University Grossman School of Medicine, New York, United States; [2]Department of Neurobiology, Howard Hughes Medical Institute, Harvard Medical School, Boston, United States; [3]Department of Neurology, University of California, San Francisco, San Francisco, United States

**Abstract** Acetylcholine is widely believed to modulate the release of dopamine in the striatum of mammals. Experiments in brain slices clearly show that synchronous activation of striatal cholinergic interneurons is sufficient to drive dopamine release via axo-axonal stimulation of nicotinic acetylcholine receptors. However, evidence for this mechanism in vivo has been less forthcoming. Mohebi, Collins and Berke recently reported that, in awake behaving rats, optogenetic activation of striatal cholinergic interneurons with blue light readily evokes dopamine release measured with the red fluorescent sensor RdLight1 (Mohebi et al., 2023). Here, we show that blue light alone alters the fluorescent properties of RdLight1 in a manner that may be misconstrued as phasic dopamine release, and that this artefactual photoactivation can account for the effects attributed to cholinergic interneurons. Our findings indicate that measurements of dopamine using the red-shifted fluorescent sensor RdLight1 should be interpreted with caution when combined with optogenetics. In light of this and other publications that did not observe large acetylcholine-evoked dopamine transients in vivo, the conditions under which such release occurs in behaving animals remain unknown.

**\*For correspondence:**
nicolas.tritsch@nyulangone.org

†These authors contributed equally to this work

**Competing interest:** The authors declare that no competing interests exist.

## Introduction

Presynaptic modulation is a ubiquitous mechanism through which neural circuits control the amount of chemical transmitter that axons release per incoming action potential (*Lovinger et al., 2022*). Striatum-projecting midbrain dopamine (DA) neurons are no exception; many transmitters present in the striatum directly act on DA axons to facilitate or depress vesicular release of DA (*Sulzer et al., 2016*). However, the axons of DA neurons do stand out in their sensitivity to acetylcholine (ACh), whose release from striatal cholinergic interneurons in brain slices is potent enough to trigger axonal action potentials and locally evoke DA release (i.e., independently of somatic spiking activity) via the activation of β2-containing nicotinic ACh receptors on DA axons (*Cachope et al., 2012*; *Threlfell et al., 2012*; *Wang et al., 2014*; *Mamaligas et al., 2016*; *Kramer et al., 2022*; *Liu et al., 2022*; *Matityahu et al., 2023*). This mechanism was proposed to underlie observations that DA release in ventral striatum (nucleus accumbens, NAc) can increase even when the firing of DA neurons in the midbrain appears unchanged (*Berke, 2018*; *Mohebi et al., 2019*). However, direct evidence for ACh-evoked striatal DA release in vivo remains limited.

In their original report of ACh-evoked DA release, *Cachope et al., 2012* presented an example recording whereby strong optogenetic activation of cholinergic interneurons for several seconds is accompanied by DA elevation in the NAc of a urethane-anesthetized mouse. More recently, we and

others showed in mice and rats that the patterns of DA and ACh release in various striatal locations in vivo are strongly correlated on sub-second time-scales in a manner that is consistent with ACh-evoked DA release (*Howe et al., 2019*; *Liu et al., 2022*; *Chantranupong et al., 2023*; *Krok et al., 2023*; *Mohebi et al., 2023*). Yet, we were unable to observe strong changes in the dynamics of DA in the dorsal and lateral striatum of mice after molecular, pharmacological and optogenetic interference with ACh signaling (*Chantranupong et al., 2023*; *Krok et al., 2023*).

A recent paper (*Mohebi et al., 2023*) provided some of the most compelling evidence to date that optogenetic activation of channelrhodopsin-expressing striatal cholinergic interneurons drives DA release in the NAc of awake behaving rats, as measured with the red-shifted DA sensor RdLight1 (*Patriarchi et al., 2020*). However, one concern with these experiments is that mApple-based fluorescent sensors – including RdLight1 and the GRAB-rDA series (*Zhuo et al., 2024*) – may exhibit photoactivation (also known as 'photoswitching' or 'photoconversion'), a process whereby mApple's red fluorescence changes in the presence of blue light (*Shaner et al., 2008*). This phenomenon is one of the main downsides of the R-GECO family of red $Ca^{2+}$ indicators, which also use mApple and grow brighter independently of $Ca^{2+}$ for hundreds of milliseconds following brief flashes of blue light, limiting their use with optogenetics (*Akerboom et al., 2013*; *Dana et al., 2016*). In the case of RdLight1, it was previously shown that photoactivation effects are negligible when expressed in cultured kidney cells and imaged using light-scanning confocal microscopy (*Patriarchi et al., 2020*). Whether RdLight1 shows photoactivation in vivo under conditions routinely used in behavioral experiments (i.e., full-field fiber photometry and optogenetic stimulation) has not been investigated.

## Results

### Blue light evokes RdLight1 fluorescence transients in the absence of an opsin

To determine if blue light modifies the fluorescent properties of RdLight1 in the behaving brain, the laboratory of N.X.T. virally expressed RdLight1 in either the dorsolateral striatum (DLS) or NAc of wild-type mice (*Figure 1A and B*) and imaged RdLight1 fluorescence in vivo using fiber photometry while mice were head-fixed on a cylindrical treadmill (*Figure 1C*). Under standard continuous illumination conditions (565 nm excitation light, 30–50 µW at the tip of the fiber), we observed transient increases and decreases in red fluorescence consistent with established patterns of DA release (*Howe and Dombeck, 2016*; *da Silva et al., 2018*; *Chantranupong et al., 2023*; *Krok et al., 2023*; *Markowitz et al., 2023*; *Mohebi et al., 2024*), including spontaneous fluctuations during immobility and large-amplitude reward-evoked responses (*Figure 1D–E*).

Delivering blue light pulses (4 ms-long) through the same fiber at powers typically used for optogenetic manipulations (9 mW at the tip of the patch cord) evoked distinct transients in RdLight1 fluorescence resembling DA release (*Figure 1E–F*). In the NAc, these transients averaged 7.0 ± 0.1% ΔF/F in magnitude, peaked 115±4ms after light onset and decayed back to baseline within 1 s ($\tau_{decay}$: 422±21ms; N=8 mice). In the DLS, blue light-evoked transients were smaller (4.5 ± 0.2% ΔF/F) but showed similar kinetics (time from light onset to peak: 129±2ms; $\tau_{decay}$: 516±60ms; N=8 mice). In both regions, blue light-evoked transients scaled in magnitude with the duration (*Figure 1G–H*) and intensity of light pulses (*Figure 1I*). In parallel experiments taking place in the laboratory of B.L.S., comparable increases in RdLight1 fluorescence were observed in N=3 mice that expressed RdLight1 in the ventrolateral striatum (VLS; *Figure 1—figure supplement 1*), confirming their occurrence across a range of experimental conditions. These delayed fluorescent signals are specific to RdLight1, as they are not observed in mice expressing the red fluorescent protein tdTomato (not shown).

### Blue light-evoked RdLight1 fluorescence transients do not reflect DA release

Do these blue light-evoked RdLight1 transients reflect a phasic elevation in extracellular DA? Several lines of evidence suggest that this is not the case. First, our wild-type mice do not express blue light-gated opsins to drive DA release and DA neurons are not thought to be intrinsically sensitive to blue light. Second, blue light-evoked transients show little trial-by-trial variability in their amplitudes and kinetics (*Figure 1G and J*). Third, blue light-evoked transients do not display short-term facilitation or depression under a variety of stimulation conditions (*Figure 2* and *Figure 1—figure supplement 1C*),

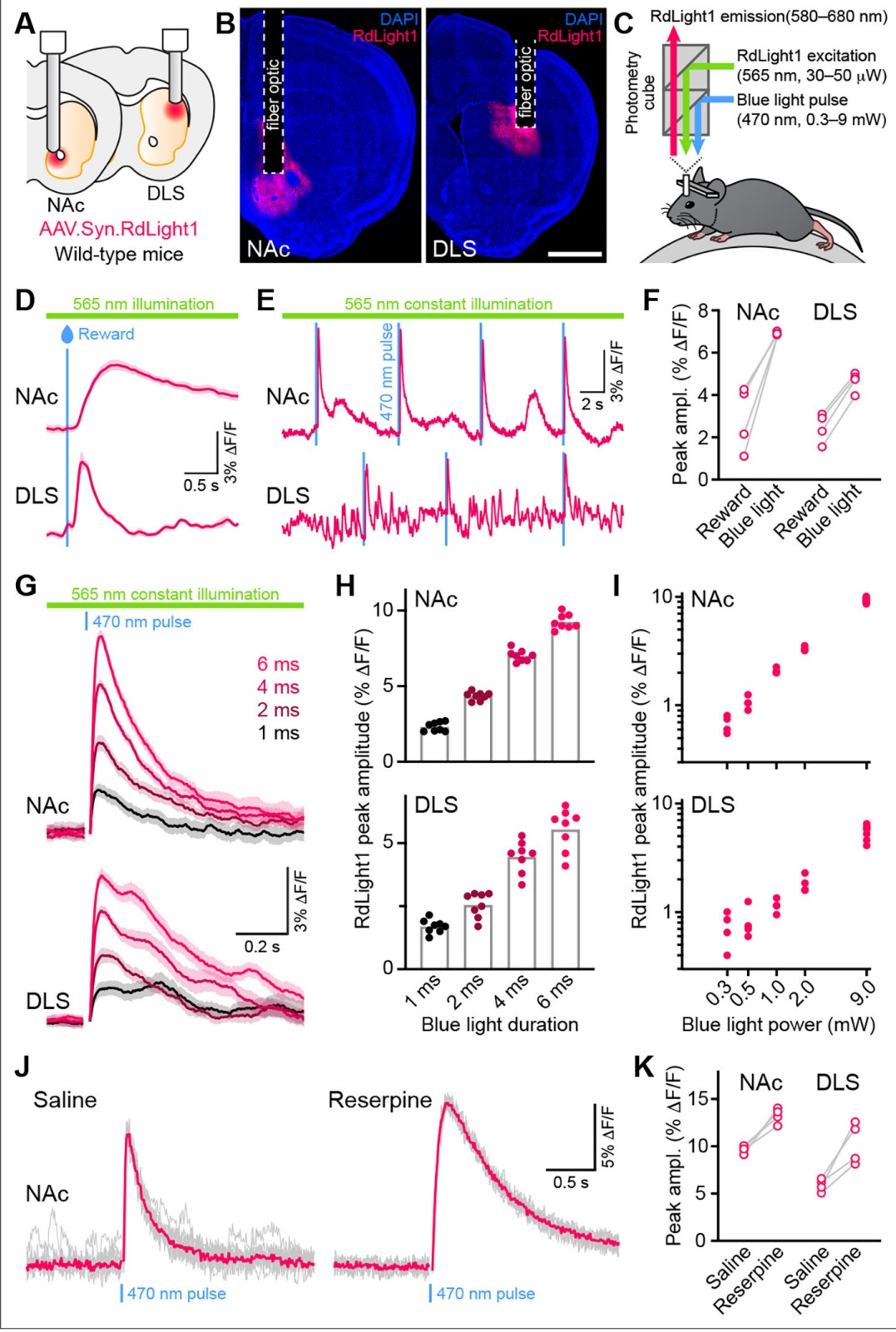

**Figure 1.** Blue light evokes RdLight1 photoactivation transients resembling DA release. (**A**) The red DA sensor RdLight1 was virally expressed in neurons of the nucleus accumbens (NAc) or dorsolateral striatum (DLS) of wild-type mice and imaged in vivo via chronically-implanted fiber optic cannulas. (**B**) Example fixed coronal sections from two mice stained for the nuclear marker DAPI (blue) and imaged by epifluorescence showing RdLight1 (red)

*Figure 1 continued on next page*

*Figure 1 continued*

expression in the NAc (left) and DLS (right). Scale bar: 1 mm. (**C**) Experimental setup for imaging RdLight1 by photometry while delivering blue light (470 nm) via the same fiber optic in awake behaving mice. RdLight1 was excited with continuous yellow-green light (565 nm) and red emitted fluorescence (580–680 nm) was collected via a dual color photometry minicube. (**D**) Representative RdLight1 transient aligned to water reward delivery in NAc (top) and DLS (bottom). Solid lines are the mean of 15 trials. Shaded area shows SEM. (**E**) Continuous RdLight1 photometric imaging during immobility in NAc (top) and DLS (bottom). Blue lines highlight 4 ms-long pulses of blue light (9 mW) delivered through the same fiber optic, each followed by a delayed RdLight1 transient. (**F**) Comparison of mean reward- and blue light-evoked RdLight1 transients recorded in N=4 mice in NAc and DLS. (**G**) RdLight1 fluorescence (emitted upon continuous excitation with 565 nm light) increases upon exposure to blue light pulses (470 nm; 9 mW at tip of patch cord) of various durations (1–6ms; color-coded) in the NAc (top) and DLS (bottom) of a representative mouse. Solid lines are the mean of 15 blue light presentations. Shaded area shows SEM. (**H**) The magnitude of blue light-evoked RdLight1 fluorescence transients (i.e., photoactivation) grows with the duration of blue light pulses in both NAc (top; N=8 mice) and DLS (bottom; N=8 mice).(**I**) Magnitude of RdLight1 fluorescence transients evoked by 6 ms-long blue light pulses of various intensities (0.3–9 mW, measured at tip of patch cord) in both NAc (top) and DLS (bottom). Data plotted on $Log_{10}$-$Log_{10}$ scales. (**J**) Representative RdLight1 photoactivation transients (red: mean of 10 individual traces shown in gray) imaged in NAc before and after systemic block of vesicular DA release with reserpine. (**K**) Magnitude of blue light-evoked (9 mW; 6ms pulse width) RdLight1 photoactivation before and after reserpine treatment in NAc (N=4 mice) and DLS (N=4 mice).

The online version of this article includes the following figure supplement(s) for figure 1:

**Figure supplement 1.** Blue light evokes RdLight1 photoactivation in VLS.

calling into question their synaptic origin. Fourth, we repeated the above experiments in a subset of mice treated with reserpine, an irreversible antagonist of the transporter required for loading DA into synaptic vesicles (***Figure 1J***). Under these conditions, spontaneous fluctuations in RdLight1 fluorescence vanished in both the DLS (N=4) and NAc (N=4), confirming the absence of activity-dependent DA release in vivo. By contrast, blue light-evoked RdLight1 transients did not disappear and, if anything, grew in amplitude and duration in both DLS and NAc (***Figure 1J and K***), demonstrating that they do not reflect synaptic release of DA.

## Discussion

Our results show that RdLight1 displays strong photoactivation following exposure to blue light under wide-field illumination conditions routinely used to monitor and manipulate neural activity in vivo. This photoactivation manifests as a prolonged, DA-independent increase in RdLight1 fluorescence that outlasts the blue light pulse and slowly decays back to baseline over hundreds of milliseconds, giving it the appearance of synaptically-released DA. Under our recording conditions, photoactivation remained detectable with as little as 0.3 mW blue light (***Figure 1I***), indicating that RdLight1 fluorescence should be interpreted with caution when combined with blue light in a variety of experimental conditions, including dual-color imaging of green and red fluorophores. In this case, the steady or rapidly-modulating (i.e., tens to hundreds of Hz) blue light might lead to constant activation of mApple but, due to its long photoswitching time, is unlikely to contribute to phasic and aperiodic transients in the red channel, which our experiments showed reflect DA release (***Chantranupong et al., 2023***; ***Krok et al., 2023***). It is possible that the initial characterization of RdLight1 failed to reveal strong blue light-mediated photoactivation as it used laser-scanning microscopy (***Patriarchi et al., 2020***), which Roger Tsien and colleagues found in their original characterization of mApple produced significantly less photoswitching compared to continuous wide-field illumination (***Shaner et al., 2008***).

Our findings call into question the nature of the RdLight1 fluorescent transients reported in ***Figure 1*** of the study by ***Mohebi et al., 2023***. Given the similarity of our recordings in terms of response magnitude, timing and dynamics over a variety of stimulation parameters, it is likely that the light-evoked RdLight1 responses reported reflect this photoactivation effect. Although the study used 405 nm illumination to control for changes in fluorophore properties independent of ligand binding (i.e., the so-called isosbestic point), this deep blue wavelength has not been shown to be the isosbestic point for RdLight1. Indeed, the isosbestic point of commonly-used green dopamine sensors is more green-shifted (e.g. 440 nm; ***Sun et al., 2020***), stressing the need for sensors to be published along with their full excitation/emission spectra, and for experimenters to tailor illumination wavelengths to their

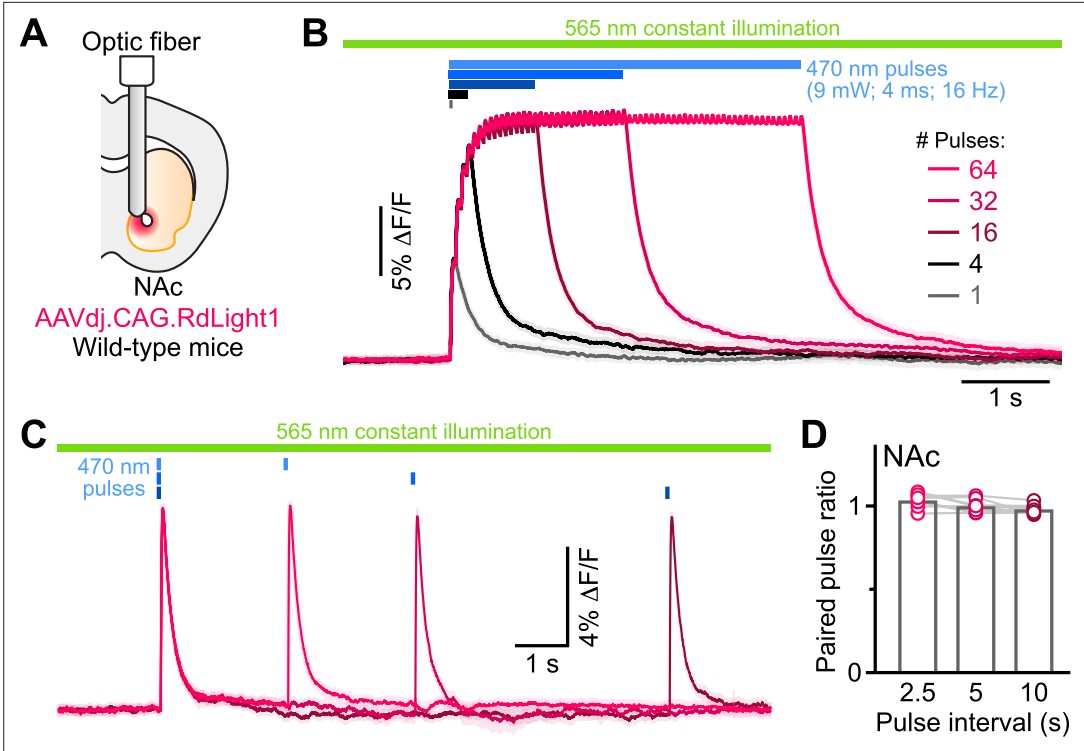

**Figure 2.** RdLight1 photoactivation transients evoked by blue light pulse trains. (**A**) Experimental setup. (**B**) Mean (± SEM) RdLight1 photoactivation in response to 1, 4, 16, 32, or 64 blue light pulses (9 mW, 4ms width, 16 Hz frequency) in the NAc of a representative mouse. Green bar illustrates constant 565 nm illumination. Blue bars show when blue light is delivered. (**C**) Same as (**B**) for pairs of blue light pulses (9 mW, 4ms width) separated by 2.5, 5 and 10 s. (**D**) Mean paired-pulse ratio (magnitude of pulse #2/pulse #1) across three inter-pulse intervals in each of N=8 mice.

specific sensor. It also remains to be determined whether illumination at isosbestic wavelengths can contribute to or modify photoactivation.

Moving forward, additional experiments will be needed to determine conditions under which cholinergic interneurons locally evoke DA release in the striatum of behaving animals. If using RdLight1, experiments should include controls to account for the effects of blue light alone, such as expressing RdLight1 only (i.e., no opsin), RdLight1 and a fluorophore [e.g., green fluorescent protein (GFP)] or a mutated sensor that does not bind DA. Blue light illumination power should be kept as low as possible. Alternatively, optogenetic experiments may be designed using GFP-based DA sensors such as dLight1 (*Patriarchi et al., 2018*) or GRAB-DA3 (*Zhuo et al., 2024*) in combination with red-shifted optogenetic actuators (*Klapoetke et al., 2014*; *Marshel et al., 2019*), although this configuration is not without caveats either, as continuous blue light illumination may cause opsin activation.

Importantly, photoactivation is not unique to RdLight1. Other mApple-based sensors, including the GRAB-rDA and R-GenGAR-DA families of red DA sensors (*Sun et al., 2020*; *Nakamoto et al., 2021*; *Zhuo et al., 2024*), as well as the R-GECO family of Ca2 +indicators (*Akerboom et al., 2013*; *Dana et al., 2016*) show photoactivation when presented with blue light, with individual constructs differing in the polarity, magnitude and duration of photoactivation effects. Photoactivation also extends beyond mApple, as many fluorophores can modify their biophysical properties when presented with different wavelengths of light. It is, for instance, the biophysical principle on which some molecular tracking and super-resolution imaging methods are founded (*Lippincott-Schwartz and Patterson, 2009*; *Chozinski et al., 2014*). Experiments should therefore be designed carefully to control for these and other potential confounds, particularly when novel sensors are introduced.

## Materials and methods

### Animals

Procedures were performed in accordance with protocols approved by the NYU Grossman School of Medicine (NYUGSM; #IA16-02082) and Harvard Medical School (HMS; #IS0000571) Institutional Animal Care and Use Committees. Wild type mice (C57BL/6 J; Jackson Laboratory strain #000664; 12–18 weeks of age) were housed in group before surgery and singly after surgery under a reverse 12 hour light-dark cycle (dark from 6 a.m. to 6 p.m. at NYUGSM and 10 a.m. to 10 p.m. at HMS) with ad libitum access to food and water.

### Stereotaxic surgery

Mice were prepared for intracranial infections of adeno-associated viruses (AAVs) as before (*Chantranupong et al., 2023*; *Krok et al., 2023*). Briefly, mice were anaesthetized with isoflurane, administered Ketoprofen (10 mg/kg, subcutaneous) or Carprofen (5 mg/kg, subcutaneous) and placed in a stereotaxic apparatus (Kopf Instruments), where a small craniotomy was drilled above the NAc (from Bregma: AP +1.0 mm, ML +0.75 mm), DLS (from Bregma: AP +0.5 mm, ML +2.5 mm), or VLS (from Bregma: AP +0.6 mm, ML +2.3 mm). 300 nL of AAV2/9.Syn.RdLight1 (CNP Viral Vector Core at the CERVO Research Center contribution) was injected (100 nL/min) at a depth of 3.7 mm below dura for NAc, 2.3 mm for DLS or 3.2 mm for VLS using a microsyringe pump (KD Scientific; Legato 111) connected to a pulled glass injection needle (100 µm tip; Drummond Wiretrol II). Fiber optics (NAc and DLS: 400 µm diameter core, 0.5 NA; RWD Life Science Inc; VLS: 200 µm diameter core, 0.48 NA; Doric) were implanted 100 µm above the injection site and cemented to the skull using C&B meta-bond (Parkell) along with a custom titanium headpost placed over lambda to allow for head fixation during recordings. Mice were allowed to recover in their home cage for 2–4 weeks before head-fixation and treadmill habituation, and recordings.

### Fiber Photometry

RdLight1 photometry recordings were carried out by feeding constant, low-power yellow-green excitation light (565 nm LED, 30–50 µW at the tip of the patch cord; Thorlabs M565F3) to a fluorescence mini cube (FMC5_E1(460–490)_F1(500–540)_E2(555–570)_F2(580–680)_S; Doric) connected to the mouse's fiber optic implant via a 0.48 NA patch cord (NAc and DLS: MFP_400/460/900–0.48_2 m_FCM-MF1.25; VLS: MFP_200/220/900_2 m_FCM-MF1.25; both from Doric). The red light emitted by RdLight1 was collected through the same patch cord and routed via the fluorescence mini cube and a second fiber optic (MFP_600/630/LWMJ-0.48_0.5 m_FCM-FCM; Doric) to a photoreceiver (Newport 2151) to produce a voltage that is proportional to the intensity of the emitted light. Voltages were digitized at 2 kHz with either a National Instruments acquisition board (NI USB-6343) or a Labjack (T7) and saved to disk as 'trials/sweeps' lasting 5–20 s in duration each using Wavesurfer software (Janelia). To characterize the photoactivation behavior of RdLight1, we delivered brief pulses (1–6ms in duration) of blue light (NAc, DLS: 470 nm LED, Thorlabs M470F3; VLS: 470 nm laser, Optoengine) to the brain via the same fluorescence mini cube and patch cord used for photometry. We tested a range of blue light powers (measured at the tip of the patch cord) frequently used for optogenetic manipulations in vivo. The timing, duration and intensity of blue light pulses were controlled digitally using Wavesurfer, with each stimulation parameter repeated at minimum 10 times per mouse/recording site. RdLight1 photoactivation responses are extremely stable over time and were reliably seen for the duration of 1 h-long recording sessions.

### Fiber Photometry Analysis

Photometry signals were processed and analyzed offline using custom code (https://github.com/TritschLab/Taniguchi-Melani-2024; copy archived at *TritschLab, 2024*) in MATLAB (Mathworks) and Igor Pro 6.02 A (Wavemetrics). Raw voltages collected from the photoreceiver were down sampled to 1,000 Hz and converted to 'percent changes in fluorescence' using the equation $\frac{\Delta F}{F} = \frac{F - F_0}{F_0}$, where $F_0$ is the mean baseline fluorescence, computed for each trial/sweep during a 1.5–2 s baseline window preceding the blue light stimulus (*Taniguchi, 2024*). Blue excitation light led to an instantaneous artifact in the red channel for the exact duration of the blue light pulse, which was blanked in display panels. For each experiment, 10–15 replicates were performed. In figures, gray traces represent single trials, while colored traces represent the mean of 10–15 trials, with standard error of the mean (SEM)

shown as a shaded area. The properties of RdLight1 photoactivation transients [peak amplitude, latency to peak (i.e., from blue light onset to RdLight1 photoactivation peak) and decay time constant (i.e., time from peak to 37% of peak)] were measured for each mouse using averaged waveforms. Data are reported in the text and figures as mean ± SEM. N-values represent the number of mice.

## Immunohistochemistry

Mice were deeply anesthetized with isoflurane and perfused transcardially with 4% paraformaldehyde in phosphate buffered saline (PBS). Brains were post-fixed for 24 h and sectioned coronally (100 μm in thickness) using a vibratome (Leica; VT1000S). Brain sections were mounted on superfrost slides and coverslipped with ProLong antifade reagent with DAPI (Molecular Probes). RdLight1 fluorescence was not immuno-enhanced. Whole sections were imaged with an Olympus VS120 slide scanning microscope.

## Reagents

To inhibit DA vesicular transport and prevent vesicular release of DA, mice were injected intraperitoneally with the irreversible vesicular monoamine transporter inhibitor reserpine (5 mg/kg) for 24 h prior to RdLight1 photometry.

## Acknowledgements

This work was supported by the National Institutes of Health (R01MH130658 to NXT), a Irma T Hirschl Research Award (to NXT), a National Science Foundation Graduate Research Fellowship (to JT) and a Howard Hughes Medical Institute Hanna Gray Fellowship (to LC, BLS). We acknowledge the New York University Langone Health Department of Comparative Medicine for animal care and maintenance and the Neuroscience Institute's imaging facilities for microscope availability.

## Additional information

### Funding

| Funder | Grant reference number | Author |
|---|---|---|
| National Institutes of Health | R01MH130658 | Nicolas X Tritsch |
| Irma T. Hirschl Trust | | Nicolas X Tritsch |
| National Science Foundation | | James Taniguchi |
| Howard Hughes Medical Institute | | Lynne Chantranupong<br>Bernardo L Sabatini |

The funders had no role in study design, data collection and interpretation, or the decision to submit the work for publication.

### Author contributions

James Taniguchi, Riccardo Melani, Lynne Chantranupong, Formal analysis, Investigation, Methodology, Writing – review and editing; Michelle J Wen, Investigation; Ali Mohebi, Joshua D Berke, Writing – review and editing; Bernardo L Sabatini, Nicolas X Tritsch, Conceptualization, Supervision, Funding acquisition, Investigation, Methodology, Writing – original draft, Project administration, Writing – review and editing

### Author ORCIDs

James Taniguchi http://orcid.org/0000-0002-1571-3869
Lynne Chantranupong http://orcid.org/0000-0001-9814-5264
Ali Mohebi http://orcid.org/0000-0001-7291-3448
Joshua D Berke http://orcid.org/0000-0003-1436-6823
Bernardo L Sabatini http://orcid.org/0000-0003-0095-9177
Nicolas X Tritsch https://orcid.org/0000-0003-3181-7681

### Ethics

This study was performed in strict accordance with the recommendations in the Guide for the Care and Use of Laboratory Animals of the National Institutes of Health. All procedures were performed in accordance with protocols approved by the NYU Grossman School of Medicine (NYUGSM; #IA16-02082) and Harvard Medical School (HMS; #IS0000571) Institutional Animal Care and Use Committees.

### Decision letter and Author response

Decision letter https://doi.org/10.7554/eLife.95694.sa1
Author response https://doi.org/10.7554/eLife.95694.sa2

## Additional files

### Supplementary files

- MDAR checklist

- Source data 1. Source data for histograms presented in *Figures 1 and 2*.

### Data availability

All data plotted in the manuscript's figures is provided in *Source data 1*.

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
