## [Decision Letter]

*In the interests of transparency, eLife includes the editorial decision letter and accompanying author responses. A lightly edited version of the letter sent to the authors after peer review is shown, indicating the most substantive concerns; minor comments are not usually included.*

Thank you for submitting your work entitled "Comment on 'Accumbens cholinergic interneurons dynamically promote dopamine release and enable motivation'" for consideration by *eLife*. Your article has been evaluated by Michael Taffe as Senior Editor, Joseph Cheer as Reviewing Editor, and two reviewers (who have opted to remain anonymous).

Please consider the comments and suggestions made by the two reviewers (please see below) and respond accordingly. Please also address the editorial issues.

*Reviewer #1:*

The comment by Taniguchi et al. systematically and convincingly demonstrates that the red dopamine sensor RdLight1 is photoactivated in response to blue light in the absence of blue light-activated opsins. Moreover, the dynamics of RdLight1 responses to blue light closely mimic the classic dynamics of dopamine transients and would be difficult to distinguish without proper controls. The magnitude of the artifact scales with blue light duration and intensity, is reproducible across several regions of striatum (and in separate laboratories), is robust to dopamine depletion with reserpine, and shows no evidence of depression with repeated stimulation. Taken together, these data highlight an important caveat for the use of RdLight1 with blue-light-activated opsins. This issue has implications for both the interpretation of the paper by Mohebi, Collins and Berke and for the wider audience of scientists utilizing fluorescent sensors in their work. While the findings of this comment show definitively that Figure 1 of Mohebi et al. is contaminated by an artifact, the remaining figures of the paper add valuable knowledge to our limited understanding of the role of cholinergic interneurons in modulating dopamine activity in the striatum in vivo. It is important to note that this comment is an example of responsible and transparent work that reflects how science should be conducted. Both the authors of the original paper and of this comment showed commendable scientific integrity in producing this comment. The fact that these studies were done across laboratories shows not only open collaboration, which should be applauded but also adds an extra layer of rigor.

1. It is still unclear if dual-color fiber photometry (as is used in some of the remaining Mohebi et al. figures) would be subject to the same artifact, as the light intensities generally used for fiber photometry are much lower than for optogenetics and lower than were tested in this comment. In the future, all authors making use of fluorescent sensors can use the approaches demonstrated in this comment to conduct proper controls for their own experiments.

2. The authors correctly point out that many mApple-based fluorescent sensors have been shown to have similar photoswitching artifacts (Zhuo et al., 2023), but should be careful about singling out mApple as the sole issue. Doing so could lead researchers to falsely assume non-mApple-based sensors are safe from such artifacts. Researchers should be reminded of the need to run appropriate controls for all new sensors.

3. The authors mention that the 405 nm wavelength has not been shown to be an appropriate isosbestic wavelength for RdLight1. This provides an opportunity to highlight the need for those developing new sensors to carefully test and publish the full excitation and emission spectra of their sensors in the future.

*Reviewer #2:*

In the present Comment Taniguchi et al. address an important methodological issue concerning the primary data collection in Mohebi et al. (2023) that substantially changes the interpretation and impact of the original article. In Mohebi et al., the authors used a red-shifted fluorescent dopamine sensor, RdLight1, to measure dopamine release in behaving rodents. They then simultaneously activated a population of acetylcholine-releasing interneurons (ChIs) using a blue-light gated channel rhodopsin (ChR2). They found that evoking action potentials in the ChIs with blue light led to transient increases in the RdLight1 signal that were interpreted as dopamine release. This result was the key observation for the principal conclusions of their study.

However, the present Comment clearly demonstrates through new data that blue light stimulation also directly 'activates' the RdLight1 sensor to produce a signal that appears as dopamine but is in fact not dopamine. Taniguchi et al. thoroughly tested and verified these findings. These results directly challenge the principal findings from Mohebi et al., including the observation that dopamine release is linearly scaled with intensity and duration of activation of cholinergic interneurons (i.e. no short-term depression).

It should be noted that Mohebi and Berke collaborated on this Comment and acknowledge the methodological error in their original publication. This Comment should be published with no need for additional data. I have no substantive concerns.

There are two changes to the text I would ask the authors to make.

1. The language used in a sentence in the Introduction implies that previous work examining the function of nicotinic receptors on dopaminergic fibers implicated these receptors as the sole drivers of all dopamine release in the dorsal and lateral striatum. This is because it is written that "DA dynamics in the dorsal and lateral striatum were found to persist even after …. interference with ACh signaling". Likely everyone would agree that the majority of "DA dynamics" should persist in these conditions. Please amend this section to use more precise language giving appropriate context to these experiments and results. For example, "previous results were unable to detect a change in DA dynamics…"

2. The authors of this Comment are experts in RdLight1, and similar sensors based on mApple. It would be a tremendous benefit to the field if the authors could leverage that expertise here to identify other sensors that could be impacted by the present results.

a. Are there other red fluorescent sensors that are based on mApple?

b. If the authors had used mApple instead of tdTomato in the experiment discussed in the Results, do they think they would have seen a signal like the one shown by RdLight1? Or is the problem with RdLight1 due to the molecular alterations necessary to make it respond to dopamine?

Please also make the following editorial revisions:

a) Abstract

Reading the article without the cover letter, it is not clear that the author list includes two authors from the paper that is being criticized (Mohebi and Berke): it would be good if this could be made clear by revising the abstract as follows:

It is widely believed that acetylcholine modulates the release of dopamine in the striatum of mammals. Experiments in brain slices clearly show that synchronous activation of striatal cholinergic interneurons is sufficient to drive dopamine release via axo-axonal stimulation of nicotinic acetylcholine receptors, but there is less evidence for this mechanism in vivo. Mohebi, Collins and Berke recently reported that, in awake behaving rats, optogenetic activation of striatal cholinergic interneurons with blue light readily evokes dopamine release, as measured with the red fluorescent sensor RdLight1 (Mohebi et al., 2023). Here, we show that blue light alone alters the fluorescent properties of RdLight1 in a manner that may be misconstrued as phasic dopamine release and that this artefactual photoactivation can account for the effects attributed to cholinergic interneurons. Measurements of dopamine using RdLight1 should, therefore, be interpreted with caution when combined with optogenetics. In light of these results (which were obtained by a multi-laboratory collaboration that included Mohebi and Berke), and the results of other studies that did not observe large acetylcholine-evoked dopamine transients in vivo, the conditions under which such release occurs in behaving animals remain unknown.

b) Results section

The statement "In a separate laboratory..." will confuse readers: please revise the Results section to make clear where the different experiments were performed.

---

## [Author Response]

Reviewer #1:The comment by Taniguchi et al. systematically and convincingly demonstrates that the red dopamine sensor RdLight1 is photoactivated in response to blue light in the absence of blue light-activated opsins. Moreover, the dynamics of RdLight1 responses to blue light closely mimic the classic dynamics of dopamine transients and would be difficult to distinguish without proper controls. The magnitude of the artifact scales with blue light duration and intensity, is reproducible across several regions of striatum (and in separate laboratories), is robust to dopamine depletion with reserpine, and shows no evidence of depression with repeated stimulation. Taken together, these data highlight an important caveat for the use of RdLight1 with blue-light-activated opsins. This issue has implications for both the interpretation of the paper by Mohebi, Collins and Berke and for the wider audience of scientists utilizing fluorescent sensors in their work. While the findings of this comment show definitively that Figure 1 of Mohebi et al. is contaminated by an artifact, the remaining figures of the paper add valuable knowledge to our limited understanding of the role of cholinergic interneurons in modulating dopamine activity in the striatum in vivo. It is important to note that this comment is an example of responsible and transparent work that reflects how science should be conducted. Both the authors of the original paper and of this comment showed commendable scientific integrity in producing this comment. The fact that these studies were done across laboratories shows not only open collaboration, which should be applauded but also adds an extra layer of rigor.1. It is still unclear if dual-color fiber photometry (as is used in some of the remaining Mohebi et al. figures) would be subject to the same artifact, as the light intensities generally used for fiber photometry are much lower than for optogenetics and lower than were tested in this comment. In the future, all authors making use of fluorescent sensors can use the approaches demonstrated in this comment to conduct proper controls for their own experiments.

Although we cannot know for sure whether the flashes of blue light used to image GCaMP6 (expressed in cholinergic interneurons) can evoke photoactivation of RdLight1 under the original experimental conditions, we do not believe they account for the large RdLight1 fluorescence transients depicted in Figures 2 and 3 of the study by Mohebi *et al.* for several reasons. First, as the Reviewer correctly points out, photometry typically uses much less power than optogenetics (<100 uW vs. >1 mW). Our data suggest that the magnitude of the photoactivation artifact scales with blue light power (see our Figure 1I). Thus, while some photoactivation may take place any time blue light is applied, it is likely exceedingly small compared to the fluorescence transients evoked by dopamine. Second, the flashes of blue light used to image GCaMP6 were delivered at a constant rate and intensity. As such, they could not account for the large and aperiodic RdLight1 fluorescence transients observed. Lastly, close inspection of the example traces reveals that many RdLight1 and GCaMP6 transients occur independently of one another, providing additional evidence that one process does not drive the other. We added a paragraph at in the discussion to that effect:

“RdLight1 fluorescence should be interpreted with caution when combined with blue light in a variety of experimental conditions, including dual-color imaging with alternating stimulation of green and red fluorophores. In this case, the steady or rapidly-modulating (i.e., tens to hundreds of Hz) blue light might lead to constant activation of mApple but, due to its long photoswitching time, is unlikely to contribute to phasic and aperiodic transients in the red channel.”

2. The authors correctly point out that many mApple-based fluorescent sensors have been shown to have similar photoswitching artifacts (Zhuo et al., 2023), but should be careful about singling out mApple as the sole issue. Doing so could lead researchers to falsely assume non-mApple-based sensors are safe from such artifacts. Researchers should be reminded of the need to run appropriate controls for all new sensors.

We thank the Reviewer for this comment. We added a paragraph at the end of the discussion to emphasize this point:

“Importantly, photoactivation is not unique to RdLight1: other mApple-based sensors, including the GRAB-rDA and R-GenGAR-DA families of red DA sensors, as well as the R-GECO family of Ca^2+^ indicators also show photoactivation when presented with blue light, with individual constructs differing in the polarity, magnitude and duration of photoactivation effects. Photoactivation also extends beyond mApple, as many fluorophores can modify their biophysical properties when presented with different wavelengths of light. It is, for instance, the biophysical principle on which some molecular tracking and super-resolution imaging methods are founded. Experiments should therefore be designed carefully to control for these and other potential confounds, particularly when novel sensors are introduced.”

3. The authors mention that the 405 nm wavelength has not been shown to be an appropriate isosbestic wavelength for RdLight1. This provides an opportunity to highlight the need for those developing new sensors to carefully test and publish the full excitation and emission spectra of their sensors in the future.

We agree with the Reviewer. We added the following sentence in the discussion:

“Indeed, even the isosbestic point of commonly used green dopamine sensors is more green-shifted (e.g. 440 nm), stressing the need for sensors to be published along with their full excitation/emission spectra, and for experimenters to tailor illumination wavelengths to their specific sensor.”

Reviewer #2:In the present Comment Taniguchi et al. address an important methodological issue concerning the primary data collection in Mohebi et al. (2023) that substantially changes the interpretation and impact of the original article. In Mohebi et al., the authors used a red-shifted fluorescent dopamine sensor, RdLight1, to measure dopamine release in behaving rodents. They then simultaneously activated a population of acetylcholine-releasing interneurons (ChIs) using a blue-light gated channel rhodopsin (ChR2). They found that evoking action potentials in the ChIs with blue light led to transient increases in the RdLight1 signal that were interpreted as dopamine release. This result was the key observation for the principal conclusions of their study.However, the present Comment clearly demonstrates through new data that blue light stimulation also directly 'activates' the RdLight1 sensor to produce a signal that appears as dopamine but is in fact not dopamine. Taniguchi et al. thoroughly tested and verified these findings. These results directly challenge the principal findings from Mohebi et al., including the observation that dopamine release is linearly scaled with intensity and duration of activation of cholinergic interneurons (i.e. no short-term depression).It should be noted that Mohebi and Berke collaborated on this Comment and acknowledge the methodological error in their original publication. This Comment should be published with no need for additional data. I have no substantive concerns.There are two changes to the text I would ask the authors to make.1. The language used in a sentence in the Introduction implies that previous work examining the function of nicotinic receptors on dopaminergic fibers implicated these receptors as the sole drivers of all dopamine release in the dorsal and lateral striatum. This is because it is written that "DA dynamics in the dorsal and lateral striatum were found to persist even after …. interference with ACh signaling". Likely everyone would agree that the majority of "DA dynamics" should persist in these conditions. Please amend this section to use more precise language giving appropriate context to these experiments and results. For example, "previous results were unable to detect a change in DA dynamics…"

We thank the Reviewer for their suggestion. We modified the highlighted sentence to read:

“Yet, we were unable to observe strong changes in the dynamics of DA in the dorsal and lateral striatum of mice after molecular, pharmacological and optogenetic interference with ACh signaling.”

2. The authors of this Comment are experts in RdLight1, and similar sensors based on mApple. It would be a tremendous benefit to the field if the authors could leverage that expertise here to identify other sensors that could be impacted by the present results.a. Are there other red fluorescent sensors that are based on mApple?

Yes. All of the red fluorescent neuromodulator sensors developed to date use circular permutated (cp) mApple, including the RdLight (Patriarchi et al., 2020), GRAB-rDA (Sun et al., 2020; Zhuo et al., 2023) and R-GenGAR-DA (Nakamoto et al., 2021) families of red DA sensors. The GRAB family of red serotonin sensors also uses cpmApple (Deng et al., 2024). We now write a sentence in the discussion to that effect:

“Importantly, photoactivation is not unique to RdLight1: other mApple-based sensors, including the GRAB-rDA and R-GenGAR-DA families of red DA sensors, as well as the R-GECO family of Ca^2+^ indicators show photoactivation when presented with blue light, with individual constructs differing in the polarity, magnitude and duration of photoactivation effects.”

b. If the authors had used mApple instead of tdTomato in the experiment discussed in the Results, do they think they would have seen a signal like the one shown by RdLight1? Or is the problem with RdLight1 due to the molecular alterations necessary to make it respond to dopamine?

Having not tried the suggested experiment, it is difficult to answer the question in the affirmative, but our hypothesis is that we would. In the original paper describing the development of mApple, Shaner, Tsien and colleagues noted that mApple displays ‘complex reversible photoswitching behavior’ that is more pronounced with continuous wide-field illumination than with laser-scanning microscopy (Shaner et al., 2008). Interestingly, their attempts at fixing this problem using mutagenesis failed. We also know that all red DA sensors developed to date (by several groups, independently) show some form of photoactivation, although individual variants differ in the polarity, magnitude and duration of such effects (see Nakamoto et al., 2021; Zhuo et al., 2023). Lastly, photoactivation also extends to mApple-based Ca^2+^ indicators (Akerboom et al., 2013; Dana et al., 2016) and to fluorophores not based on mApple (Patterson and Lippincott-Schwartz 2002; Lippincott-Schwartz and Patterson 2009). Photoactivation is therefore neither specific to RdLight1, nor to the molecular alterations necessary to turn mApple into a fluorescent reporter for dopamine (i.e. circular permutation of mApple and insertion into one of the intracellular loops of a G protein-coupled DA receptor).

References cited in this response

Akerboom J, Carreras Calderon N, Tian L, Wabnig S, Prigge M, Tolo J, Gordus A, Orger MB, et al. (2013). Genetically encoded calcium indicators for multi-color neural activity imaging and combination with optogenetics. Front Mol Neurosci **6**: 2.

Dana H, Mohar B, Sun Y, Narayan S, Gordus A, Hasseman JP, Tsegaye G, Holt GT, et al. (2016). Sensitive red protein calcium indicators for imaging neural activity. *ELife*
**5**: e12727.

Deng F, Wan J, Li G, Dong H, Xia X, Wang Y, Li X, Zhuang C, et al. (2024). Improved green and red GRAB sensors for monitoring spatiotemporal serotonin release in vivo. Nat Methods DOI: 10.1038/s41592-024-02188-8.

Lippincott-Schwartz J and Patterson GH (2009). Photoactivatable fluorescent proteins for diffraction-limited and super-resolution imaging. Trends Cell Biol **19**(11): 555-565.

Nakamoto C, Goto Y, Tomizawa Y, Fukata Y, Fukata M, Harpsoe K, Gloriam DE, Aoki K and Takeuchi T (2021). A novel red fluorescence dopamine biosensor selectively detects dopamine in the presence of norepinephrine in vitro. Mol Brain **14**(1): 173.

Patriarchi T, Mohebi A, Sun J, Marley A, Liang R, Dong C, Puhger K, Mizuno GO, et al. (2020). An expanded palette of dopamine sensors for multiplex imaging in vivo. Nat Methods **17**(11): 1147-1155.

Patterson GH and Lippincott-Schwartz J (2002). A photoactivatable GFP for selective photolabeling of proteins and cells. Science **297**(5588): 1873-1877.

Shaner NC, Lin MZ, Mckeown MR, Steinbach PA, Hazelwood KL, Davidson MW and Tsien RY (2008). Improving the photostability of bright monomeric orange and red fluorescent proteins. Nat Methods **5**(6): 545-551.

Sun F, Zhou J, Dai B, Qian T, Zeng J, Li X, Zhuo Y, Zhang Y, et al. (2020). Next-generation GRAB sensors for monitoring dopaminergic activity in vivo. Nat Methods **17**(11): 1156-1166.

Zhuo Y, Luo B, Yi X, Dong H, Miao X, Wan J, Williams JT, Campbell MG, et al. (2023). Improved green and red GRAB sensors for monitoring dopaminergic activity in vivo. Nat Methods DOI: 10.1038/s41592-023-02100-w.

Please also make the following editorial revisions:a) AbstractReading the article without the cover letter, it is not clear that the author list includes two authors from the paper that is being criticized (Mohebi and Berke): it would be good if this could be made clear by revising the abstract as follows:It is widely believed that acetylcholine modulates the release of dopamine in the striatum of mammals. Experiments in brain slices clearly show that synchronous activation of striatal cholinergic interneurons is sufficient to drive dopamine release via axo-axonal stimulation of nicotinic acetylcholine receptors, but there is less evidence for this mechanism in vivo. Mohebi, Collins and Berke recently reported that, in awake behaving rats, optogenetic activation of striatal cholinergic interneurons with blue light readily evokes dopamine release, as measured with the red fluorescent sensor RdLight1 (Mohebi et al., 2023). Here, we show that blue light alone alters the fluorescent properties of RdLight1 in a manner that may be misconstrued as phasic dopamine release and that this artefactual photoactivation can account for the effects attributed to cholinergic interneurons. Measurements of dopamine using RdLight1 should, therefore, be interpreted with caution when combined with optogenetics. In light of these results (which were obtained by a multi-laboratory collaboration that included Mohebi and Berke), and the results of other studies that did not observe large acetylcholine-evoked dopamine transients in vivo, the conditions under which such release occurs in behaving animals remain unknown.

Thank you for this suggestion. In consultation with the authors, we agreed to implemented the first suggestion (“Mohebi, Collins and Berke recently reported that…”) but not the second (“In light of these results (which were obtained by a multi-laboratory collaboration that included Mohebi and Berke)…”) for the reason that Drs. Mohebi and Berke did not contribute to data collection. We did, however, include a new paragraph in the discussion describing how the various laboratories came together to produce this manuscript.

b) Results sectionThe statement "In a separate laboratory..." will confuse readers: please revise the Results section to make clear where the different experiments were performed.

To clarify where the different experiments were performed, we now mention in the result section whether results were obtained in the laboratory of N.X.T. or B.L.S.